

# Genome-wide identification and expression analysis of ADP-ribosylation factors associated with biotic and abiotic stress in wheat (*Triticum aestivum* L.)

Yaqian Li[1,2], Jinghan Song[1,2], Guang Zhu[1], Zehao Hou[2], Lin Wang[2], Xiaoxue Wu[2], Zhengwu Fang[2], Yike Liu[1,2] and Chunbao Gao[1,2]

[1] Hubei Key Laboratory of Food Crop Germplasm and Genetic Improvement, Food Crops Institute, Hubei Academy of Agricultural Sciences/Hubei Engineering and Technology Research Center of Wheat/Wheat Disease Biology Research Station for Central China, Wuhan, China

[2] Engineering Research Center of Ecology and Agricultural Use of Wetland, Ministry of Education/Hubei Collaborative Innovation Center for Grain Industry/College of Agriculture, Yangtze University, Jingzhou, China

Corresponding author
Yike Liu, hbliuyk@foxmail.com

## ABSTRACT

The *ARF* gene family plays important roles in intracellular transport in eukaryotes and is involved in conferring tolerance to biotic and abiotic stresses in plants. To explore the role of these genes in the development of wheat (Triticum aestivum L.), 74 wheat *ARF* genes (TaARFs; including 18 alternate transcripts) were identified and clustered into seven sub-groups. Phylogenetic analysis revealed that TaARFA1 sub-group genes were strongly conserved. Numerous *cis*-elements functionally associated with the stress response and hormones were identified in the TaARFA1 sub-group, implying that these *TaARFs* are induced in response to abiotic and biotic stresses in wheat. According to available transcriptome data and qRT-PCR analysis, the *TaARFA1* genes displayed tissue-specific expression patterns and were regulated by biotic stress (powdery mildew and stripe rust) and abiotic stress (cold, heat, ABA, drought and NaCl). Protein interaction network analysis further indicated that TaARFA1 proteins may interact with protein phosphatase 2C (PP2C), which is a key protein in the ABA signaling pathway. This comprehensive analysis will be useful for further functional characterization of *TaARF* genes and the development of high-quality wheat varieties.

## INTRODUCTION

Intracellular transport is crucial for cell survival and growth, and promotes the formation of cell membranes and lysosomes to enable the secretion of proteins, hormones, and neurotransmitters, and absorb exogenous molecules by endocytosis. The transmembrane transport of macromolecules and granular substances in eukaryotic cells is carried out by encapsulating them in vesicles with a lipid bilayer. Vesicle transport is carefully controlled by regulatory molecules, including ADP-ribosylation factors (ARFs) (*Boman et al., 2000*). ARFs are allosteric activators of cholera toxin, and are in an activated state when bound to GTP, and in a deactivated state when bound to GDP (*Wonderlich et al., 2011*; *Bourgoin,*

*2012*). *ARFs* belong to the *Ras* superfamily, which is divided into five subfamilies: *Ras*, *Rho*, *Rab*, *Ran,* and *ARF* (*Burgoyne, 2001*). The *ARF* subfamily lacks the C-terminal isoprenylation and carboxymethylation regions, which is different from other members of the *Ras* superfamily, but possesses an additional nucleotide-sensitive region, an extended N-terminus, and a covalently attached myristate, which complement each other to constitute a 'myristoyl switch' (*Bérauddufour et al., 1999*). GTP hydrolysis dissociates ARF, which then coats proteins on the membrane so that the vesicles can dock and fuse with target membranes (*Gebbie et al., 2005*). Their molecular mass is approximately 21 kDa, and the *ARF* gene family were classified into *ARFs* and *ARF-like* (*ARL*) genes based on their amino acid (aa) sequence homology (*Bourgoin, 2012*; *Muthamilarasan et al., 2016*). ARF proteins are highly conserved (>60% similarity among themselves) and have similar biological activities, while ARL proteins are highly divergent (40–60% identity) and play roles in different pathways, including secretory pathways (*Li et al., 2004*).

The primary structure of cholera toxin is highly conserved, and the toxin plays important roles in intracellular transport in eukaryotes (*Mi Hee et al., 2002*). For example, in *Arabidopsis*, the absence of *ARF1* in the Golgi apparatus results in the development of an abnormal structure, inhibition of protein transport, and inhibition of plant growth and development (*Myung Ki et al., 2013*). *ARFs* also play roles in resisting biotic and abiotic stresses in plants (*Muthamilarasan et al., 2016*; *Lee et al., 2003*). *ARFs* in rice may be related to stress resistance, and *ARF1* might participate in various plant defense signaling pathways (*Lee et al., 2003*). Overexpression of ARF1 confers salt and drought tolerance in rice, *Arabadopsis* transgenic plants, and *Spartina alterniflora*. (*Joshi et al., 2014*).

Common wheat (*Triticum aestivum* L.) is one of the most important crops for humans, and occupies a large production area, providing staple food globally (*Veraverbeke & Delcour, 2002*; *Shiferaw et al., 2013*). However, the quality of wheat is often affected by adverse environmental conditions, including drought, salt stress, heavy metals, low temperature, pests, and diseases (*Bajwa, Farooq & Nawaz, 2018*). Considering the important roles of *ARFs* in plant growth and development, and response to abiotic and biotic stresses, a comprehensive understanding of *ARFs* in wheat (*TaARFs*) would contribute to functional understanding of *ARFs* and support resistance breeding. In the present study, Chinese Spring genome data (IWGSC RefSeq v1.1) was used to conduct a systematic and comprehensive phylogenetic analysis of the *TaARFs*, including the gene structures, conserved motifs, chromosomal locations, *cis*-elements, synteny, and duplication patterns of ARF protein sequences. The STRING database was used to generate protein-protein networks between TaARF proteins so as to better understand gene functions. Furthermore, based on the available transcriptome data and qRT-PCR analysis, we analyzed the expression profiles of *TaARFs* at different development stages and under different abiotic and biotic stresses to predict their potential functions and regulatory patterns, which provides a theory evidence for wheat stress resistance gene breeding.

## MATERIALS & METHODS

### Identification of ARF genes

Computer-based methods were used to identify members of the *ARF* gene family from the wheat reference genome IWGSC RefSeq v1.1 assembly (https://wheat-urgi.versailles.inra.fr). Known ARF protein sequences, including 19 ARFs from *Arabidopsis thaliana* (AtARFs) and 21 ARFs from *Oryza sativa* (OsARFs), were used as query sequences for BLASTp analysis with an e-value cutoff of $<1 \times 10^{-10}$ (*Altschul et al., 1997*). The obtained sequences were submitted to InterProScan (http://www.ebi.ac.uk/interpro/) to check the ARF domains *Mulder & Apweiler (2007)*. The Pfam database (http://pfam.xfam.org/) was used to select sequences that contained the ARF-box domain (PF00025) (*Finn et al., 2006*). The hit sequences were further validated using SMART (http://smart.embl-heidelberg.de/smart) to remove unmatched proteins (*Letunic et al., 2004*).

### Phylogenetic analysis and characterization of TaARFs

A total of 96 protein sequences (19 from *Arabidopsis* (*Gebbie et al., 2005*), 21 from rice (*Muthamilarasan et al., 2016*), and 56 from wheat) were compared by ClustalW2 software with default parameters (*Oliver et al., 2005*). An unrooted phylogenetic tree was created using the Neighbor-Joining (NJ) method with 1000 replicated-bootstraps in MEGA 7.0 software (*Kumar, Stecher & Tamura, 2016*). Then, the phylogenetic tree was edited by the Interactive Tree of Life (ITOL, Version 3.2.317, http://itol.embl.de/) to have the final illustration (*Letunic, 2016*). The identified TaARFs were used to perform protein characterization in ExPASy Server10 (https://prosite.expasy.org/) (*Wilkins et al., 1999*). The predicted protein features for each of the protein sequences were determined, including the length, molecular weight (MW), instability index, isoelectric point (pI), and aliphatic index. Plant-mPLoc (http://www.csbio.sjtu.edu.cn/bioinf/plant-multi/) was used to predict the cellular localization (*Horton et al., 2006*).

### Exon-intron structure and motif analysis of *TaARFs*

Structural analysis was performed to identify the exon-intron structure of each *TaARF* gene using GSDS (http://gsds.cbi.pku.edu.cn/index.php/) (*Hu et al., 2015*). The software MEME (http://meme-suite.org/) was used to determine conserved TaARF motifs (*Bailey et al., 2006*). Motifs with *E*-values >0.001 were probably statistical artefacts rather than real motifs, and were excluded (*Bailey et al., 2006*). The parameters were employed as the following descriptions: the maximum number of motifs, 10; and the optimum width of each motif, between 6 and 50 residues. The motif prediction results were inserted into TBtools (https://github.com/CJ-Chen/TBtools/) to produce the illustrations (*Chen et al., 2018*). The locations of conserved domains were determined by SMART and visualized in MEME to reveal the diversification of *TaARFs* (*Fang et al., 2020*).

### Chromosomal locations and synteny of *TaARFs*

The location information of *TaARF* genes were obtained from the reference genome in the IWGSC v1.1 database. Moreover, we generated chromosome locations using MapInspect version 1.0. The common tool "all against all BLAST search" was used to determine possible

paralogous or orthologous sequences among wheat sequences with an $E$-value cutoff of 1e−10 and identity >80% (*Gu et al., 2002*). The R package "circlize" was employed to prepare a diagram showing the locations and homology relationships of *TaARFs* (*Zuguang et al., 2014*). The non-synonymous (Ka) and synonymous (Ks) substitution rates were calculated using DNA Sequence Polymorphism (DnaSP) 5.10 to analyze gene duplication events (*Rozas, 2009*). We calculated the Ka/Ks ratios for the replicated *ARF* gene pairs in wheat to explore whether Darwinian positive selection pressure has affected the functional proportion of replicated genes. We also compared the gene duplication events of *ARF* genes between wheat and *Triticum dicoccoides*, *Aegilops tauschii*, *Arabidopsis*, and rice. In general, when the ratio is greater than 1, the replicated gene is under positive selection, a ratio equal to 1 indicates genes under neutral evolution, and a ratio of less than 1 indicates genes under negative selection pressure (*Zhang et al., 2006*; *Anton, Makova & Li, 2002*).

## Multiple conditional transcriptome analysis of *TaARFs*

Multiple RNA-seq data from different tissues, development stages, and treatments were downloaded from the NCBI Short Read Archive (SRA) database and mapped to the wheat genome using HISAT2. Cufflinks were used to perform gene assembly, expression level calculations, and identifications of differences in differentially expressed genes (*Cole et al., 2012*). The obtained transcripts per million (TPM) values reflecting the expression level of each gene were used to generate a heatmap of TaARFs using the R package "pheatmap" (*Kolde, 2015*). Triad expression analysis was carried out as described previously (*Ramírez-González et al., 2018*).

## Plant materials and qRT-PCR

Hexaploid wheat (cultivar Emai 170; *Triticum aestivum*; AABBDD) was used to validate the expression patterns of selected candidate genes in all experiments. Seeds were surface sterilized with 1% hydrogen peroxide, germinated in an incubator at 28 °C for 2 d, and transferred to a greenhouse at 26 °C with a 16 h light/8 h dark cycle and relative humidity of 60–70%. Seven-day-old seedlings were treated with cold, ABA, and NaCl. Control seedlings continued to grow under standard conditions. Root tissue was collected 6 h and 12 h after treatment, including the set controls. All sample materials were quickly frozen in liquid nitrogen and stored at −80 °C before RNA extraction. Total RNA was isolated using the Trizol reagent (Invitrogen, Carlsbad, CA, USA). Quantitative real time (RT)-PCR was performed using the TaKaRa qRT-PCR system (RR047A). The gene-specific primers (Table S10) were designed using Primer 5.0 to amplify 80–350 bp fragments. The thermal cycling conditions were as follows: 95 °C for 30 s and 40 cycles of 95 °C for 5 s and 60 °C for 30 s. The relative quantity of target gene transcripts was calculated using the $2^{-\Delta\Delta ct}$ method with wheat $\beta$-actin and GAPDH as reference genes (*Yin et al., 2018*).

## *Cis*-element analysis of putative promoter regions

The *cis*-elements in the promoter regions are related to gene expression patterns and functions (*Anne-Laure, Adrien & Veitia, 2014*). To investigate the *cis*-elements in the promoter regions of genes of interest, we downloaded 1.5 kb of the genomic DNA sequences upstream of the start codon corresponding to each gene from the hexaploid wheat database.
The Plant CARE database (http://bioinformatics.psb.ugent.be/webtools/plantcare/html/) were used to analysis the putative *cis*-elements in the promoter sequences (*Magali et al., 2002*).

### Gene Ontology annotation and protein interaction network

A GO (Gene Ontology) database was used for functional annotation of the ARF genes using MAJORBIO CLOUD (https://cloud.majorbio.com/). GO annotations were mapped according to biological processes, molecular functions, and cellular components. TaARFs protein-protein interaction networks was assembled using the STRING tool (https://string-db.org/). All predicted TaARF proteins have been submitted to the STRING database. The minimum required interaction score was set to high confidence (0.900). The active interaction sources was setted come from 'experiments' and 'databases'. The maximum number of interactors were no more than 5 on the first shell.

## RESULTS

### Identification and classification of ARF genes in wheat

In the present study, the entire hexaploid wheat genome was downloaded and used to construct a local database. After genomic retrieval, a total of 126 proteins that were similar to ARF were obtained from hexaploid wheat; however, only 77 wheat sequences were confirmed to be conserved in the ARF family domain (PF00025 and IPR006689) by Pfam and InterProScan. Sequences without complete conserved domains or with a length less than 170 aa were excluded. After elimination, 74 ARF proteins containing complete domains were obtained. Among the remaining ARF proteins, 18 were splice variants (Table S1). To investigate their evolutionary relationships, we constructed an NJ tree with MEGA7.0 using the amino acid sequences of putative *ARF* family members from *Arabidopsis*, rice, and wheat (Fig. 1, Table S2). The predicted TaARF proteins were classified into seven sub-groups: ARFA1, ARFB1, ARFC1, ARF3, ARL1, TTN5, and GB1 based on the phylogenetic tree and previous reports (*Muthamilarasan et al., 2016*). The ARL1 sub-group is the largest of these sub-groups with 17 members. The ARFA1 and the ARFB1 sub-groups have 14 and nine members, respectively. There are three proteins in the ARF3 and ARFC1 sub-groups, and the other sub-groups each contain five members. Interestingly, *Arabidopsis* had one ARFD1 protein, while rice and wheat did not have any. These results are consistent with the coevolutionary relationships between these species, indicating that the relationship between wheat and rice is closer than that with *Arabidopsis*. Each *TaARF* gene was named based on its phylogenetic relationship with *AtARFs* and *OsARFs* (*Muthamilarasan et al., 2016*). Genes corresponded equally across the three homoeologous subgenomes (A, B, and D) in wheat, which is referred to as a triad (*Ramírez-González et al., 2018*). We identified 15 *TaARFs* triads with reference to the results of *Ramírez-González et al. (2018)* (Table S3). *TaARFs* triads have identical gene names except for the sub-genome identifier (A, B or D).

### Features of predicted TaARF proteins

Of the 56 putative TaARF proteins, the predicted MW was around 21 kDa on average, except for TaARFA1a-D and TaARFB1c-B. The number of exons and introns were similar
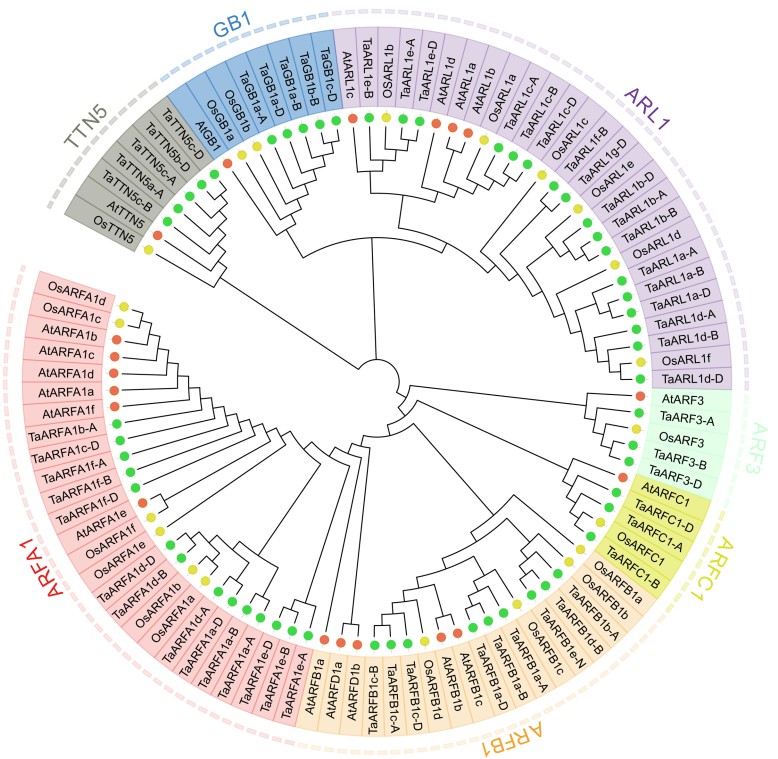

**Figure 1  Phylogenetic relationship of wheat, rice, and Arabidopsis.** Full length amino acid sequences of hexaploid wheat (Ta), Rice (Os), and Arabidopsis (At) were aligned by ClustalW2 and an unrooted neighbor-joining tree was constructed with 1,000 bootstrap iterations. The tree classified the proteins into seven distinct classes, shaded in red, orange, yellow, green, purple, blue, and grey. Proteins from rice, Arabidopsis, and wheat are represented in yellow, red, and green closed circles, respectively.

within the same class but different between classes. The protein lengths ranged from 170 (TaGB1b-B) to 263 (TaARFA1a-D) amino acids (aa) and the predicted isoelectric point ranged from 5.67 to 9.35 (Table 1). The protein instability index shows that 82% TaARFs are stable proteins, but TaARFA1a-D, TaARFB1b-A, TaARFB1c-B, TaARFB1d-B, TaARFB1e-N, TaGB1a-A, TaGB1a-B, TaGB1a-D, TaGB1b-B, and TaGB1c-D were predicted to be unstable. The average hydropathicity (GRAVY) was less than 0, except in the case of TaARFC1-A/B/D, indicating that most of these proteins are hydrophilic (Table S1). Subcellular localization prediction showed that TaARFs are localized mainly in the cytoplasm, but are also found in the chloroplasts, peroxisome, nucleus, or mitochondria (Table S1).

## Gene structure and motif analysis of the TaARF gene family

The structure of genes can be used to predict their expression and function. We found that *TaARFs* contained different exon-intron composition patterns by comparing the gene structures (Fig. 2). There were 38 *TaARF* sequences with both untranslated regions (UTRs), and of the remaining sequences, 14 did not have 5′- and 3′-UTRs, 2 *(TaARFA1d-A and TaARFA1c-D)* had only 5′-UTRs, and 2 *(TaARFA1a-D and TaARFB1b-A)* had only
**Table 1  ARF transcription factor families in wheat.**

| Gene Name | Accession numbers | Location | Exons | Introns | Length | MW | pI |
|-----------|-------------------|----------|-------|---------|--------|-----|-----|
| TaARFA1a-A | TraesCS1A02G306200.1 | chr1A:498451494-498454124 | 5 | 4 | 186 | 21.32 | 6.92 |
| TaARFA1a-B | TraesCS1B02G317000.1 | chr1B:541466983-541469854 | 6 | 5 | 184 | 21.11 | 6.97 |
| TaARFA1a-D | TraesCS1D02G305900.1 | chr1D:403235691-403239527 | 6 | 6 | 263 | 29.64 | 9.35 |
| TaARFA1b-A | TraesCS2A02G235500.1 | chr2A:293934939-293967206 | 6 | 5 | 181 | 20.57 | 6.43 |
| TaARFA1c-D | TraesCS2D02G244900.1 | chr2D:277672480-277681787 | 5 | 6 | 181 | 20.58 | 6.43 |
| TaARFA1d-A | TraesCS3A02G337300.1 | chr3A:584275968-584279016 | 6 | 7 | 182 | 20.88 | 6.43 |
| TaARFA1d-B | TraesCS3B02G368600.1 | chr3B:581026448-581031825 | 6 | 5 | 181 | 20.71 | 6.43 |
| TaARFA1d-D | TraesCS3D02G330500.2 | chr3D:443258251-443265348 | 6 | 5 | 181 | 20.71 | 6.43 |
| TaARFA1e-A | TraesCS5A02G142100.1 | chr5A:315494544-315497231 | 5 | 5 | 181 | 20.44 | 6.09 |
| TaARFA1e-B | TraesCS5B02G140900.1 | chr5B:266003993-266006657 | 5 | 5 | 181 | 20.44 | 6.09 |
| TaARFA1e-D | TraesCS5D02G150000.1 | chr5D:238710757-238712733 | 5 | 4 | 183 | 20.65 | 6.09 |
| TaARFA1f-A | TraesCS5A02G467400.1 | chr5A:645260990-645263277 | 6 | 5 | 211 | 24.02 | 7.77 |
| TaARFA1f-B | TraesCS5B02G479100.1 | chr5B:650545995-650548824 | 5 | 5 | 211 | 24.02 | 6.02 |
| TaARFA1f-D | TraesCS5D02G480200.1 | chr5D:517840384-517842860 | 6 | 5 | 211 | 23.99 | 7.77 |
| TaARFB1a-A | TraesCS1A02G197800.1 | chr1A:355714313-355716842 | 6 | 5 | 191 | 21.55 | 6.3 |
| TaARFB1a-B | TraesCS1B02G212500.1 | chr1B:386159117-386161564 | 6 | 5 | 191 | 21.55 | 6.3 |
| TaARFB1a-D | TraesCS1D02G201300.1 | chr1D:284239314-284241950 | 6 | 5 | 191 | 21.55 | 6.3 |
| TaARFB1b-A | TraesCS6A02G086900.1 | chr6A:55551759-55555137 | 6 | 6 | 194 | 21.82 | 5.58 |
| TaARFB1c-A | TraesCS6A02G268000.1 | chr6A:494440084-494443530 | 6 | 5 | 191 | 21.37 | 6.83 |
| TaARFB1d-B | TraesCS6B02G114900.1 | chr6B:98857984-98861271 | 5 | 5 | 172 | 19.45 | 5.81 |
| TaARFB1c-B | TraesCS6B02G295200.1 | chr6B:530127311-530130936 | 6 | 5 | 246 | 27.97 | 9.3 |
| TaARFB1c-D | TraesCS6D02G246900.1 | chr6D:349986601-349990113 | 6 | 5 | 191 | 21.40 | 6.83 |
| TaARFB1e-N | TraesCSU02G019600.1 | chrUn:21195216-21198738 | 6 | 6 | 194 | 21.84 | 5.82 |
| TaARFC1-A | TraesCS4A02G149200.1 | chr4A:284629181-284631736 | 2 | 1 | 184 | 20.38 | 5.94 |
| TaARFC1-B | TraesCS4B02G164400.1 | chr4B:340990997-340999096 | 2 | 2 | 184 | 20.31 | 5.84 |
| TaARF3-A | TraesCS7A02G284000.1 | chr7A:319587234-319598812 | 8 | 7 | 182 | 20.31 | 5.73 |
| TaARF3-B | TraesCS7B02G181400.1 | chr7B:274309305-274319048 | 8 | 7 | 198 | 22.29 | 6.14 |
| TaARF3-D | TraesCS7D02G282400.1 | chr7D:287897903-287906900 | 8 | 7 | 198 | 22.29 | 5.46 |
| TaARFC1-D | TraesCS4D02G152600.1 | chr4D:185795832-185798286 | 2 | 2 | 184 | 20.35 | 6.05 |
| TaARL1a-A | TraesCS3A02G172900.1 | chr3A:189275871-189280985 | 3 | 2 | 193 | 21.94 | 5.95 |
| TaARL1a-B | TraesCS3B02G203800.1 | chr3B:237643721-237648261 | 3 | 2 | 193 | 21.94 | 5.95 |
| TaARL1a-D | TraesCS3D02G179300.1 | chr3D:160908142-160912382 | 3 | 2 | 201 | 22.86 | 6.32 |
| TaARL1b-A | TraesCS3A02G193800.1 | chr3A:266450899-266454547 | 3 | 2 | 193 | 22.11 | 6.97 |
| TaARL1b-B | TraesCS3B02G221700.1 | chr3B:276848921-276852339 | 3 | 2 | 193 | 22.11 | 6.97 |
| TaARL1b-D | TraesCS3D02G196200.1 | chr3D:193594920-193598371 | 3 | 2 | 193 | 22.13 | 6.97 |
| TaARL1c-A | TraesCS4A02G045000.1 | chr4A:37062663-37067415 | 5 | 4 | 184 | 20.73 | 9.15 |
| TaARL1c-B | TraesCS4B02G260300.1 | chr4B:527276119-527280600 | 5 | 4 | 184 | 20.73 | 9.15 |
| TaARL1c-D | TraesCS4D02G260100.1 | chr4D:429465149-429469539 | 5 | 4 | 184 | 20.73 | 9.15 |
| TaARL1d-A | TraesCS5A02G089200.1 | chr5A:119438999-119442370 | 3 | 2 | 193 | 22.04 | 6.91 |
| TaARL1d-B | TraesCS5B02G095200.1 | chr5B:125228094-125232113 | 3 | 2 | 193 | 22.08 | 7.75 |
| TaARL1d-D | TraesCS5D02G101300.1 | chr5D:114349156-114352814 | 3 | 2 | 193 | 22.06 | 6.91 |

**Table 1** (*continued*)

| Gene Name | Accession numbers | Location | Exons | Introns | Length | MW | pI |
|-----------|-------------------|----------|-------|---------|--------|-----|-----|
| *TaARL1e-A* | TraesCS6A02G293900.1 | chr6A:525759292-525762110 | 6 | 5 | 184 | 20.60 | 8.32 |
| *TaARL1e-B* | TraesCS6B02G324400.1 | chr6B:573689158-573692125 | 6 | 6 | 175 | 19.49 | 8.31 |
| *TaARL1e-D* | TraesCS6D02G274800.1 | chr6D:383544182-383547096 | 6 | 6 | 184 | 20.60 | 8.32 |
| *TaARL1f-B* | TraesCS7B02G236300.1 | chr7B:440347979-440349242 | 3 | 2 | 193 | 22.14 | 6.59 |
| *TaARL1g-D* | TraesCS7D02G332100.1 | chr7D:423583936-423585168 | 3 | 2 | 193 | 22.11 | 6.21 |
| *TaGB1a-A* | TraesCS1A02G069400.1 | chr1A:52056090-52060042 | 6 | 6 | 203 | 22.93 | 5.67 |
| *TaGB1b-B* | TraesCS2B02G191100.1 | chr2B:166468400-166471957 | 6 | 4 | 170 | 19.34 | 5.89 |
| *TaGB1c-D* | TraesCS2D02G172000.1 | chr2D:115947943-115950782 | 7 | 5 | 199 | 22.51 | 7.09 |
| *TaGB1a-B* | TraesCS1B02G087700.1 | chr1B:83206061-83209210 | 6 | 6 | 202 | 22.81 | 6.07 |
| *TaGB1a-D* | TraesCS1D02G072000.1 | chr1D:52358164-52361912 | 6 | 6 | 203 | 22.94 | 5.67 |
| *TaTTN5a-A* | TraesCS1A02G421700.1 | chr1A:577760728-577762543 | 6 | 5 | 197 | 22.62 | 8.36 |
| *TaTTN5b-D* | TraesCS1D02G429700.1 | chr1D:481330712-481333368 | 5 | 5 | 185 | 21.17 | 8.46 |
| *TaTTN5c-A* | TraesCS5A02G014900.1 | chr5A:10476109-10479944 | 5 | 5 | 185 | 21.14 | 8.46 |
| *TaTTN5c-B* | TraesCS5B02G013100.1 | chr5B:13174286-13178149 | 5 | 5 | 185 | 21.08 | 8.46 |
| *TaTTN5c-D* | TraesCS5D02G020800.1 | chr5D:13715586-13718281 | 5 | 4 | 185 | 21.14 | 8.46 |

**Notes.**

Length, protein length aa; MW, molecular weight, kDa; pI, isoelectric point.

3′-UTRs (Fig. 2B). The number of introns ranged from 1 to 7, and 43% contained 6 exons of varying lengths. *TaARF* genes in the same sub-group shared similar exon-intron structure. The majority of the ARFB1 sub-group contained 6 exons, with the exception *TaARFB1d-B*, which contains five. While the ARFA1 subgroup gene members have 5 exons, ARFB1, ARFC1, and ARF3 subgroups have 6, 2 and 8 exons, respectively.

The 10 most statistically significant motifs were chosen to describe the motif pattern in TaARFs, which were named motif 1-10 (Fig. 2C, Table S4). The lengths of the 10 motifs were between 8 (motif 8 and 9) and 49 (motif 1) aa residues. The number of motifs in each TaARF protein varied from 5 to 8. Notably, motif analysis indicated that most TaARFs have relatively conserved motif compositions. All TaARF proteins (except TaGB1c-D and TaARFB1d-B) contained motif 1 to 4, which contained the ARF-box domain. In general, many TaARFs of the same group encoded proteins with similar motif compositions, and therefore, these proteins may have similar functions. There were 4 motifs (motif 1, 3, 4, and 5) in the middle region of most TaARF proteins; however, different motifs were found at the N-terminal and C-terminal regions. For example, two specific motifs (motif 9 and motif 10) were only found in the C-terminus of the TaARL1 sub-group. Motif 7 appeared in both the N-terminus and the C-terminus of the TaARL1 subfamily, but only in the C-terminus of other subfamilies.

## Chromosomal distribution and gene duplication events of TaARF genes

To further investigate the genetic differences in the *TaARF* gene family, we mapped their chromosomal locations. After positioning, 55 *TaARFs* were mapped on all 21 chromosomes, while *TaARFB1e-N* was not mapped on any chromosomes (Fig. 3A, Table 1). *TaARFs* were distributed roughly evenly across the three subgenomes (subgenome A, 18; subgenome

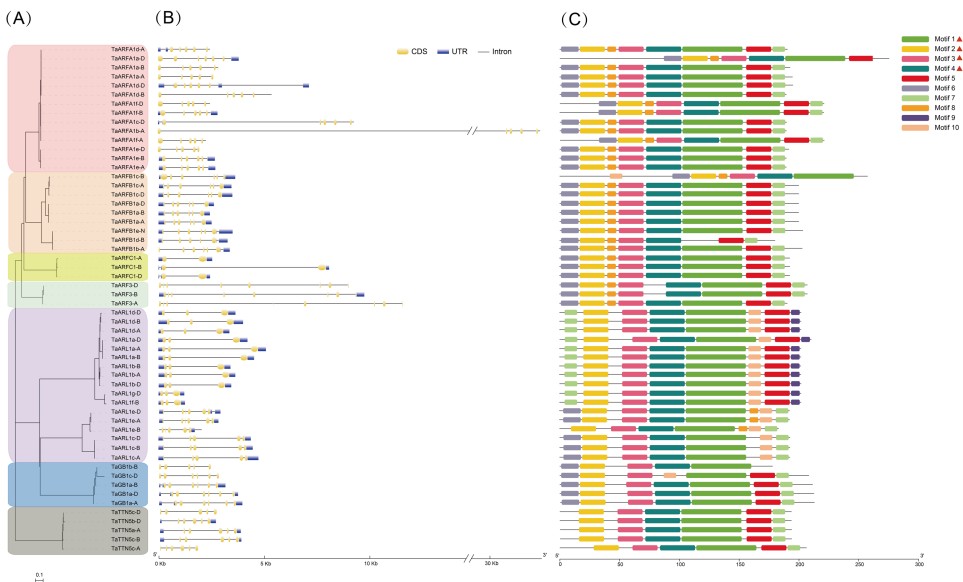

**Figure 2** **Comparative analysis of the phylogenetics, exon-intron structures, and conserved motifs of the ARF family in wheat (TaARFs).** (A) The phylogenetic tree of 56 ARF proteins; (B) gene structures of 56 ARF proteins in hexaploid wheat. The yellow boxes are coding sequences (CDSs), the black lines are introns, and the blue boxes are 5′-untranslated regions (UTRs) or 3′-UTRs; (C) Motif composition models of 56 ARF proteins. Different motifs are color-coded according to the legend.

B, 18; subgenome D, 19). Members from the same sub-group tended to be distributed at similar locations. However, the distribution of genes on chromosomes varied from one homoeologous group to another. The largest homoeologous group of 5 chromosomes (A, B, and D) had the most *TaARF* genes (12), followed by the smallest homoeologous groups of 2 (4 genes) and 7 (5 genes). By screening the sequence identity and position, 75 segmental duplication pairs were predicted, with no tandem duplication pairs being identified. Triads were not considered when predicting gene duplication events. The Ka/Ks ratio between *TaARF* gene pairs was less than 1 (average 0.08) in all cases (Fig. 3B, Table S5), suggesting that the *TaARF* gene family might have experienced strong purifying selective pressure (*Song et al., 2019*). Among the *TaARF* genes, 75 pairs of segmental duplication genes were found concentrated in the ARFA1 subgroup (Table S5).

Four comparative syntenic maps of wheat associated with four representative species, including *Triticum dicoccoides*, *Arabidopsis*, *Aegilops tauschii*, and rice, were constructed to further deduce the evolutionary origin and orthologous relationship of the wheat ARF family (Fig. 4). The numbers of orthologous pairs between the other four species (*Triticum dicoccoides*, *Aegilops tauschii*, *Arabidopsis*, and rice) were 61, 27, 67, and 88, respectively (Table S6). Six *TaARF* genes have both orthologous genes in four species. Among them, the ARFA1 subgroup accounted for 5 genes, which was a larger number than the other subfamilies.

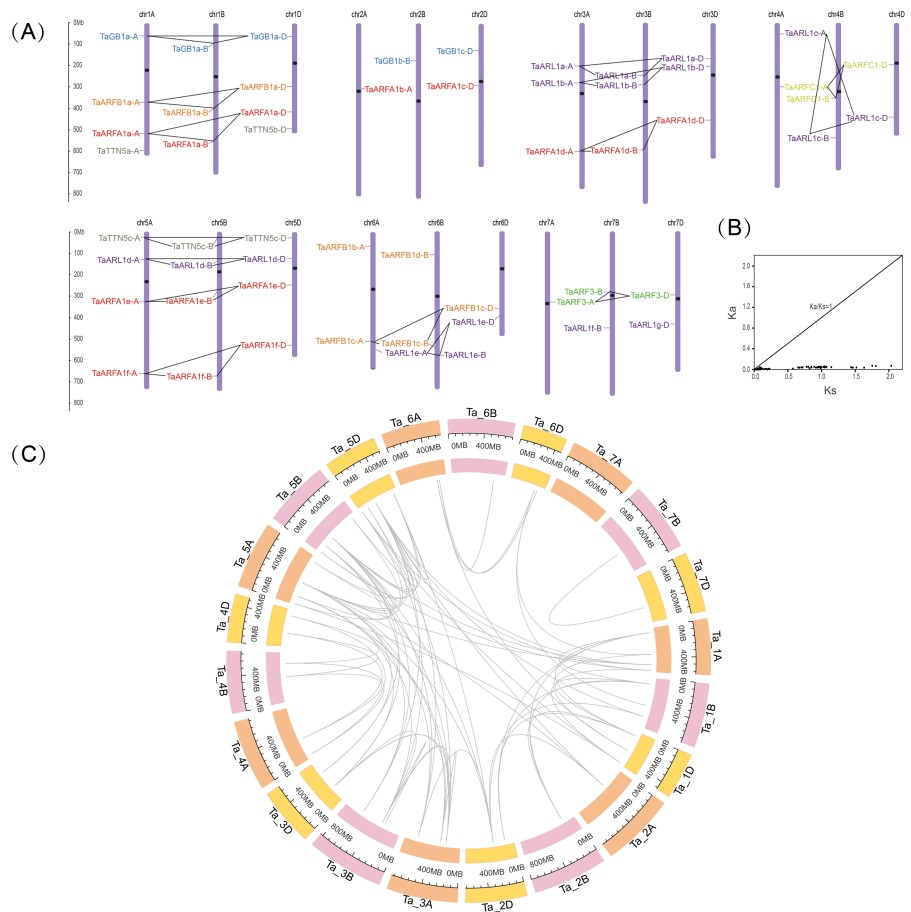

**Figure 3** **Chromosomal locations and gene duplication events in *TaARFs*.** (A) Chromosomal locations of the *TaARF* genes in wheat. The ruler on the left indicates the physical map distance between genes (Mb). The black dots represent the centromeres. Different groups of *TaARFs* are represented by different colors. The triads are indicated with gray lines; (B) Ka/Ks values for duplicated *TaARF* gene pairs; (C) gene duplication events of *TaARFs*.

## Expression analysis of TaARF genes in different tissues

RNA-sequencing (RNA-seq) is a powerful tool to explore transcription patterns using high-throughput sequencing (*Wang, Gerstein & Snyder, 2009*). The RNA-seq data for five tissues (grain, spike, leaf, stem, and root) in wheat were used to characterize the expression of *TaARF* genes during growth and development. Out of the 56 full-length genes, 93% were expressed in at least one developmental stage, with a wide expression range between 1-852 TPM (TPM$_{max}$) (Fig. 5A; Table S1, Table S7). The remaining 7% of full-length genes had a very low level of expression with a TPM$_{max} < 1$, and were considered as "not expressed" (Fig. S1, Table S1). In general, the expression of *TaARFs* could be divided into three patterns: the first group contains members that are widely expressed in many tissues during multiple developmental stages, the second group contains those that are highly induced only at specific growth and development stages, and the last group contained members with no expression or low expression during all growth and developmental

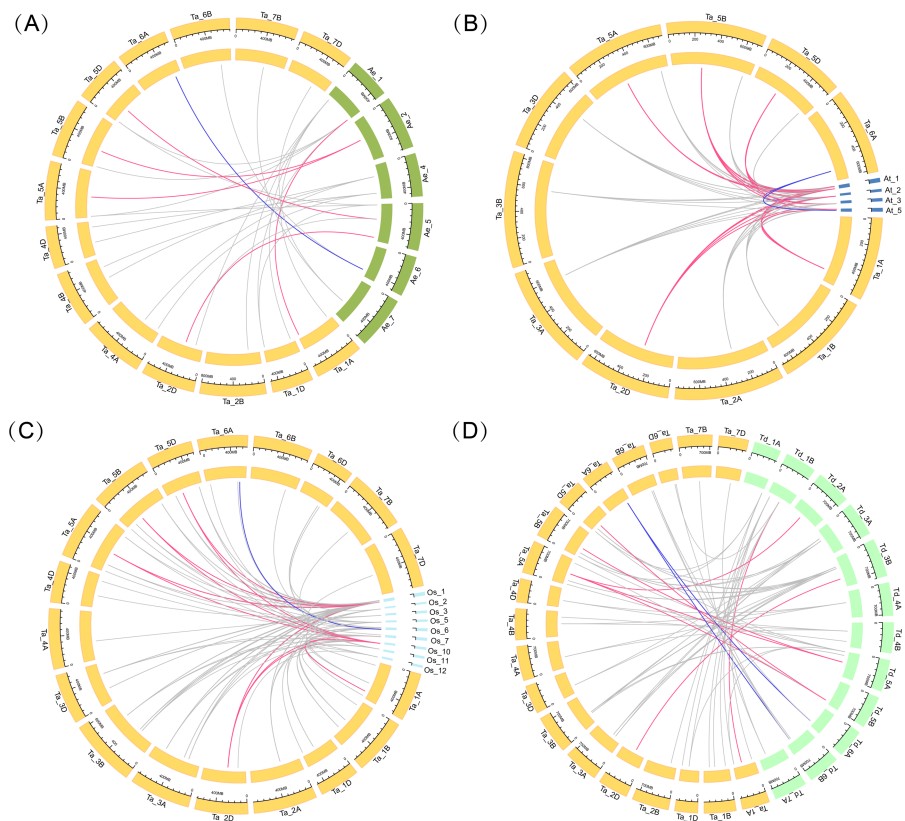

**Figure 4** **Gene duplication events of *ARF* genes between wheat and other plant species.** Gray lines in the background indicate the syntenic *ARF* gene pairs. Colored lines in the background indicate the orthologous genes across the four species, with red indicating the ARFA1 sup-group genes and blue indicating the ARL1 sup-group genes. (A) Gene duplication events of *ARF* genes between wheat and Aegilops tauschii. (B) Gene duplication events of *ARF* genes between wheat and Arabidopsis. (C) Gene duplication events of *ARF* genes between wheat and rice. (D) Gene duplication events of *ARF* genes between wheat and Triticum dicoccoides.

phases. Almost all *ARF* genes in the sub-group ARFA1 were highly expressed in multiple tissues, and may therefore be involved in the regulation of growth and development.

The differential abundance of homoeologs was analyzed using a previously described framework (*Ramírez-González et al., 2018*). Balanced expression of homoeolog triads was denoted when transcripts from every gene had a similar abundance. Suppressed and dominant categories were denoted if some transcripts were more abundant than others (Fig. 5B, Fig. S2). Expression data were obtained from a developmental time course of Chinese Spring wheat (*Ramírez-González et al., 2018*). The percentage of triads in the balanced category was between 60% and 73%, with an average of 68.3% (Table S8), which is consist with the values observed for transcripts from all wheat genes (*Ramírez-González et al., 2018*). BAR software was used to display the electron fluorescence diagram of *TaARFA1a-A* expression to better understand *TaARF* gene expression during growth and development. Our results suggest that some *TaARFs* may play an important role during wheat growth (Fig. 5C).

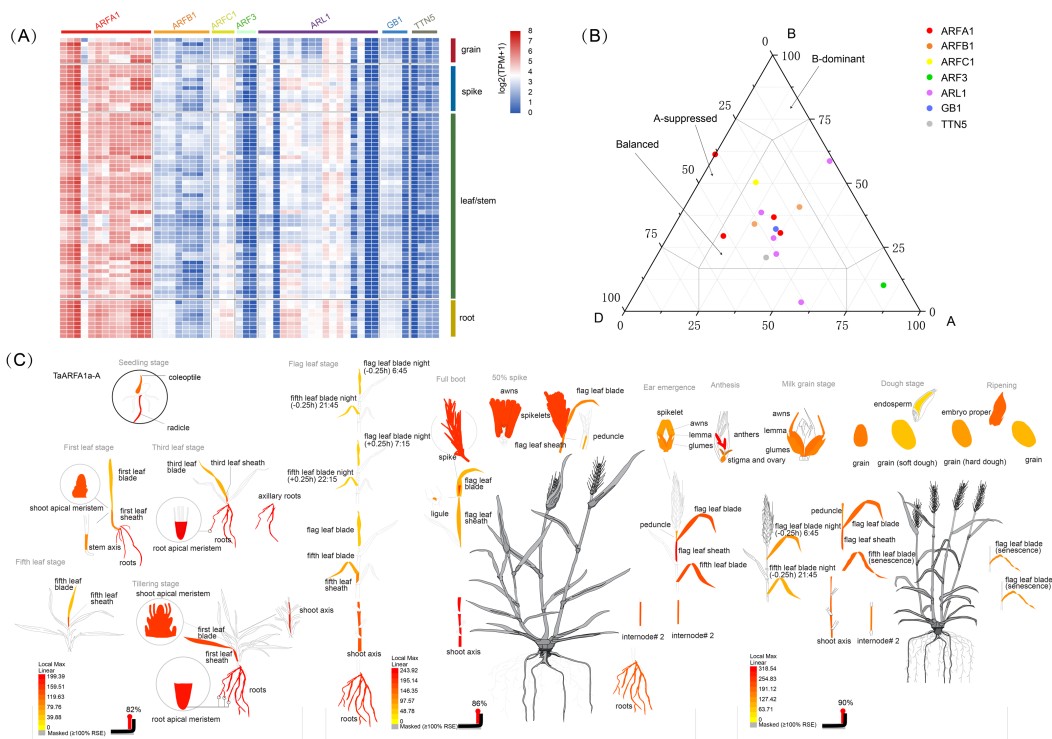

**Figure 5** **Expression analysis of *TaARF* genes in different tissues.** (A) Heatmap showing the expression levels of *TaARF* genes in different subfamilies (columns) and wheat developmental stages/tissues (rows); (B) expression balance for all 1:1:1 triads was plotted in a triangular plot with the coordinates of each circle representing the normalized expression of A, B and D homoeologs. Triads are indicated by circles, with areas separated by gray lines indicating expression patterns that are balanced, dominant for one subgenome homoeolog, or suppressed for one homoeolog, as previously described. Colored circles represent sub-groups; (C) an "electronic fluorescence pictogram" representation of the target *TaARFA1a-A* gene expression pattern based on the Wheat Wheat Atlas dataset (http://bar.utoronto.ca/efp_wheat/), modified from http://bar.utoronto.ca/efp_wheat/cgi- bin/efpWeb.cgi.

## Expression analysis of TaARF genes under biotic and abiotic stresses

The TaARFA1 sub-group members that had higher expression during different developmental stages were selected for further analysis. The original RNA-seq data related to abiotic stress (drought and heat) and biotic stress (powdery mildew and stripe rust) from the NCBI database was used for expression profiling of *TaARFA1* sub-group members (Table S9). The levels of *TaARFA1* genes were up-regulated after 6 h of drought treatments compared with the control, and then down-regulated with prolonged exposure to stress (Fig. 6A). The pattern of expression was opposite under high temperature conditions, with down-regulation followed by up-regulation to near-control levels after 12 h. With drought and heat co-treatment, *TaARFA1* genes showed similar expression patterns to under high temperature alone, although the magnitude of down-regulation was greater with multiple stressors. During biological stress, the expression levels of most *TaARFA1* sub-group genes fluctuated. Powdery mildew infection caused the expression of *TaARFA1* genes (except *TaARFA1e-A/B/D*) to be up-regulated compared to the control. The expression level of

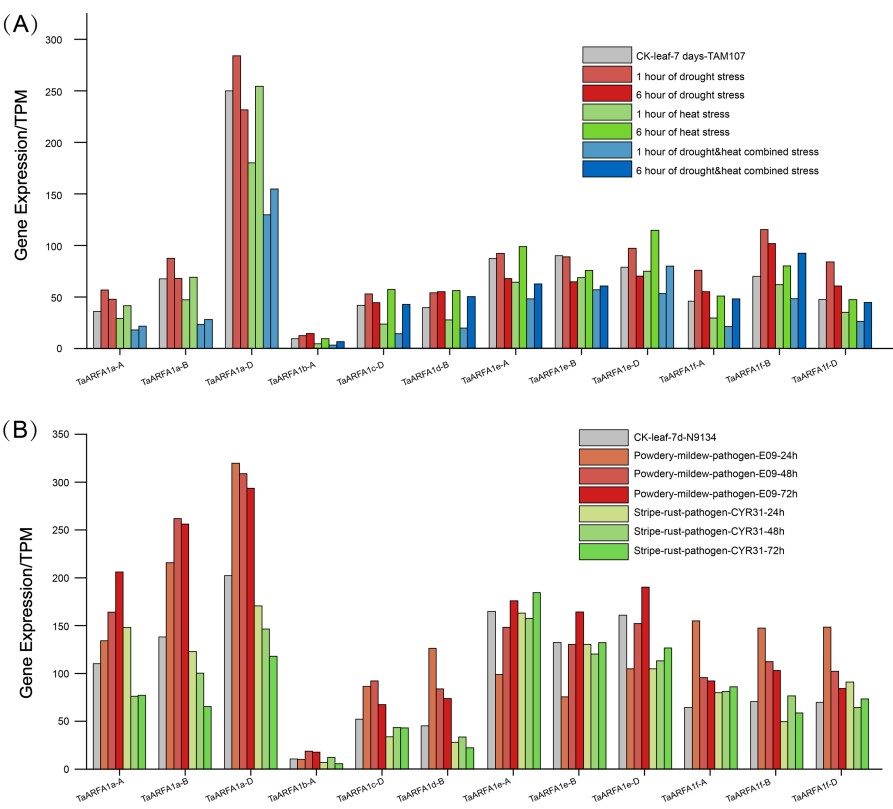

**Figure 6** Expression patterns of *ARF* genes in wheat (*TaARFs*) under different abiotic and biotic stresses. (A) Expression of TaARFA1 members in the leaf under drought, heat, and drought & heat combined stress. (B) Expression of TaARFA1 members in the leaf under powdery mildew and stripe rust stress.

*TaARFA1e-A/B/D* were down-regulated after infection for 24 h, then up-regulated with prolonged treatment, exceeding that of the control after 72 h (Fig. 6B). The magnitude of changes in expression during stripe rust infection were much smaller than during powdery mildew infection.

Due to the lack of transcriptome data for ABA and osmotic stress in the roots, we selected *TaARFA1* genes for qRT-PCR analysis under cold, salt, and ABA stress (Tables S10, S11). The qRT-PCR results revealed that *TaARFA1* genes were responsive to all abiotic stress treatments, and their expression patterns varied based on the stress type. Under ABA treatment, the expression levels of most *TaARFA1* genes were down-regulated compared with the control, except the expression levels of *TaARFA1b-A*, *TaARFA1c-D*, and *TaARFA1e-A/B/D*, which increased after 6 h of treatment but recovered to the control level after 12 h. The expression levels of *TaARFA1c-D*, *TaARFA1d-A/B/D*, and *TaARFA1e-A/B* were significantly up-regulated during NaCl and cold treatment, while *TaARFA1f-A/B/D* were down-regulated (Fig. 7).

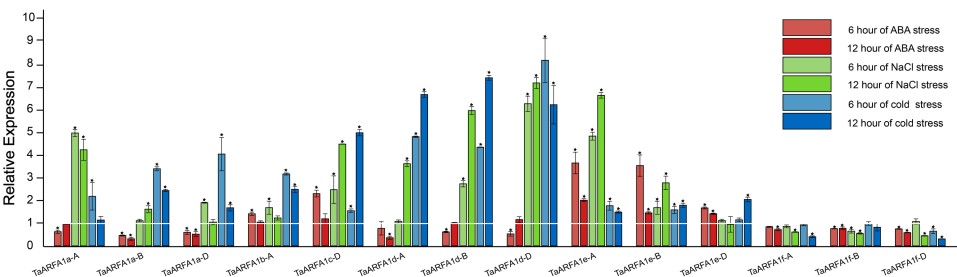

**Figure 7 Expression of TaARFA1 members in root under ABA, salt and cold stress.** The roots were sampled after 6 and 12 h of ABA (100 μM), NaCl (150 mmol), and cold (4 °C) treatments. The white line represent the expression levels of control (*, $p < 0.05$).

## *Cis*-acting elements in the promoter of TaARF

The distribution of different *cis*-acting elements in gene promoters may reflect differences in function and regulation. Identified *cis*-acting elements were divided into three major categories, namely those relating to growth and development, phytohormone response, and biotic/abiotic stress (Tables S12, S13). Among them, the CAAT-box and TATA-box were the most frequently observed, and related to growth and development. We predicted 3 *cis*-acting elements related to biotic/abiotic stress (ARE, G-box, and Sp1) and phytohormone response (ABRE, CGTCA motif, and TGACG motif) in *TaARFA1* genes (Fig. 8A). The *cis*-acting elements related to growth and development were enriched in *TaARFA1* genes (Fig. 8B).

## Gene ontology annotation and protein interaction network of TaARF genes

Gene ontology (GO) annotation is currently one of the most important functional annotation methods. In the present dataset, GO terms related to: (1) biological processes (BP), including regulation of biological process (GO: 0050789), response to stimulus (GO: 0050896), single-organism process (GO: 0044699), cellular process (GO: 0009987), signaling (GO: 0023052), biological regulation (GO: 0065007); (2) cellular components (CC), including cell (GO:0005623), organelle (GO:0043226), cell part (GO:0044464); and (3) molecular function (MF), including binding(GO:0005488), were specifically enriched (Fig. 9A, Table S14). In this study, we also used string website to predict the protein interaction network in which the *TaARF* genes were involved (Fig. S3). Proteins that share similar functions or participate in the same pathway tend to show interaction networks, so gene clusters or modules were formed in the network of protein interactions. It's obviously that TaARFA1 and PP2C proteins may have interaction relationship (Fig. 9B). PP2C is a key protein in the ABA signaling pathway; in the absence of ABA, PP2Cs were negatively regulated by repressors that suppress gene transcription (*Nguyen, Jung & Cheong, 2019*). *TaARFA1* genes might interact with PP2C to respond to biotic and abiotic stress.

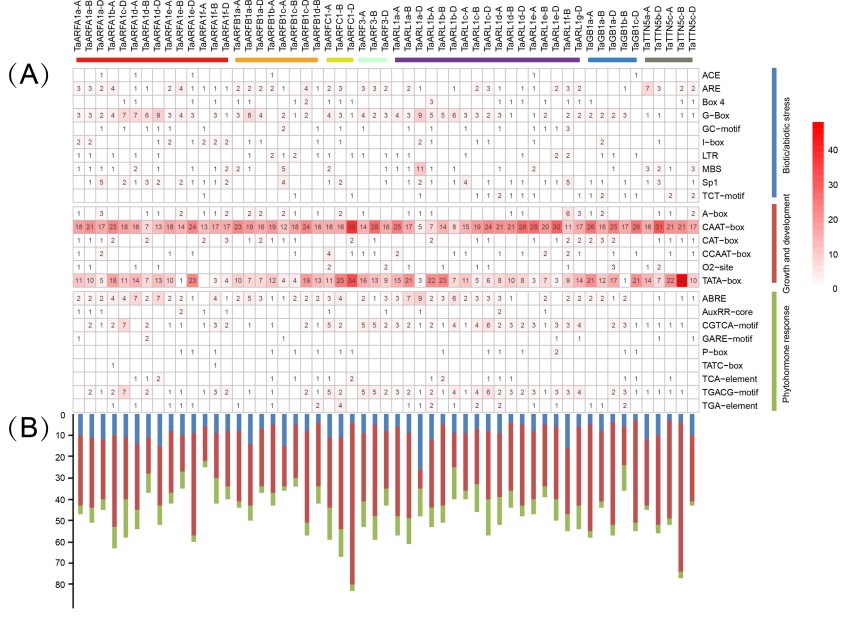

**Figure 8** **Predicted *cis*-acting elements in the promoters of *TaARF* genes.** (A) *Cis*-acting elements involved in stress in the promoter of *TaARF* genes. The different colors and numbers of the grid indicate the number of different promoter elements; (B) The number of cis-acting elements in *TaARF* genes. The histogram represents the sum of the cis-acting elements in each category.

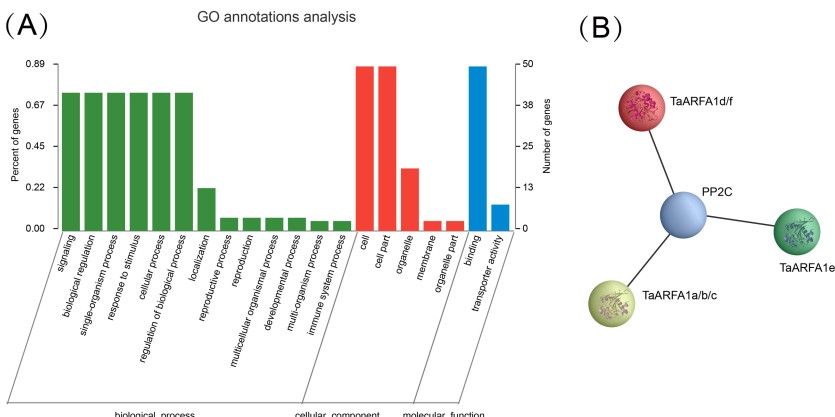

**Figure 9** **Gene ontology annotation and protein interaction network of *TaARF* genes.** (A) Gene Ontology (GO) enrichment analysis for annotated *TaARF* genes. The GO terms are grouped into three main categories, green for biological processes, red for cellular components, and blue for molecular function; (B) TaARFA1 protein interaction network.

## DISCUSSION

Many plant genomes have been analyzed during the continuous maturation of sequencing technology, allowing the identification of gene families at the whole genome level (*Takayuki et al., 2005*; *Wang et al., 2018*). ARFs function in diverse physiological and molecular
activities, and recent evidence has demonstrated their involvement in conferring tolerance to biotic and abiotic stresses in plants (*Muthamilarasan et al., 2016*). A genome-wide analysis of the *TaARF* gene family was performed previously, and 56, 30, 13, 19, and 21 ARF members were identified in hexaploid wheat, *Triticum dicoccoides*, *Aegilops tauschii*, *Arabidopsis thaliana,* and rice, respectively (*Muthamilarasan et al., 2016*). The number of *TaARF* genes is higher than other plants, possibly because wheat is a heterologous hexaploid, which has experienced two expansions during its evolution. Most genes belonging to triads in *TaARF* genes were located at similar positions on homoeologous chromosomes, supporting the theory that polyploidization events may have played a role in the expansion of *TaARF* numbers (*Zengcui et al., 2011*). In the process of polyploidization, a large number of *TaARF* genes may be created by gene replication and the conservation of *TaARF* genes. Multiple copies of *ARF* genes may have been retained in wheat because they endow plants with abiotic and biological resistance.

Duplication is also an important evolutionary process in gene family expansion, and duplicated genetic material provides opportunities for functional differentiation (*Jiang et al., 2019*; *Santoshkumar et al., 2013*). During evolution, duplicated gene pairs can experience functional divergence, contributing to the formation of new gene functions, which is essential for environmental adaptability and speciation (*Conant & Wolfe, 2008*; *Victoria & Bryan, 2002*). The analysis of gene duplication could help us to understand the evolution of genes and species. According to the Holub, a chromosomal region containing two or more copies of the same gene within 200 kb is defined as a tandem duplication event, or a segmental duplication event (*Holub, 2001*; *Schlueter et al., 2007*). In the present study, all gene duplications were segmental duplication and most segmentally duplicated gene pairs were from the same group, and located at similar positions in homoeologous chromosomes. In addition, duplicated genes were concentrated in the ARFA1 sub-group. The Ka/Ks ratios of wheat ARF genes were less than 1 (average 0.08), implying that all duplicated gene pairs were under negative selection pressure. Many of the *TaARFs* in segmentally duplicated pairs exhibited similar exon structures. We also constructed four comparative syntenic maps of wheat associated with four representative species to further deduce the evolutionary origin and orthologous relationships of the *TaARF* gene family. Among the orthologous genes, six *TaARF* genes have both orthologous genes in four species. Among them, the ARFA1 subgroup accounts for 5 genes, suggesting that *ARFA1* subgroup genes may play a crucial role. These results also show that these genes originate from an ancestral gene, and may have been generated by gene replication, which is more evolutionarily conservative and therefore more commonly observed in genes that are essential for survival. In plants possessing multiple copies of *ARF* genes, the protein content or mRNA expression would be consequently increased, possibly resulting in increased resistance to stress. Some studies have shown that increased gene dosage is beneficial to plant resistance, for example, plants with higher copy numbers of glyphosate resistance genes have stronger resistance to glyphosate (*Widholm et al., 2001*). In general, the *TaARF* gene family was constrained by evolution to maintain its functional stability.

RNA-seq analysis provides insights into the expression patterns of genes in different development stages and under a variety of stress conditions (*Ramírez-González et al.,*

*2018*). In the present study, we analyzed *TaARFA1* expression in different plant organs and determined that this subgroup had relatively high transcript accumulation, which supports their direct or indirect involvement in certain developmental stages. *TaARFA1b-A* and *TaARFB1a-A/B/D* were relatively highly expressed in root tissue. Interestingly, there are 4 *TaARF* genes that expressed at very low levels, indicating that they may not play a role in wheat growth and development. The ARF gene family can also respond to biotic and abiotic stresses in plants (*Muthamilarasan et al., 2016*). Plants are subjected to many stresses, causing survival pressure that can reduce wheat yield and negatively impact social-economic stability (*Chen et al., 2015*; *Nussbaumer et al., 2015*). Analysis of RNA-seq data revealed that most *TaARFs* had a similar expression profile and were either significantly up- or down-regulated under the tested stress conditions, supporting the fact that some of these environmental stresses share similar regulatory responses and signal transduction pathways (*Ma, 2007*). These genes may potentially play shared roles in stress resistance.

We selected *TaARFA1* genes in the root and performed qRT-PCR to validate the expression pattern that we had determined using publicly available RNA-seq data. Perhaps unsurprisingly, we found that different *TaARF* genes responded to different stresses with different expression patterns, similarly to the RNA-seq data. The gene expression patterns under different abiotic stresses were distinct, implying that the signaling pathways involved in these responses are stress-specific (*Zhao et al., 2018*). The expression levels of *TaARFA1d-A/D* in the root decreased with ABA treatment, but increased with NaCl and cold treatment. This specific stress response warrants further investigation to identify the underlying molecular mechanisms.

By predicting the interaction network of wheat ARF protein, we found that all ARFA1 members interact with PP2C, which participates in plant growth and development and also plays major roles in the response to biotic and abiotic stresses, including bacterial pathogens (*Ivy et al., 2010*), salt (*Manabe et al., 2007*), drought *Shinozaki & Yamaguchi-Shinozaki, 2006*, and abscisic acid (ABA) (*Meyer, Leube & Grill, 1994*). *TaPP2C-a10* transgenic *Arabidopsis* exhibited decreased tolerance to drought stress (*Yu et al., 2020*). Overexpression of maize *ZmPP2C* in *Arabidopsis* decreased ABA sensitivity and plant drought tolerance (*Liu et al., 2009*). PP2Cs are abscisic acid (ABA) co-receptors that negatively regulate the ABA signaling pathway by inhibiting downstream SnRK2 protein kinases (*Sreenivasuluab et al., 2012*). ARFA1 may therefore suppress the expression of *PP2C* genes.

## CONCLUSIONS

In the present study, a total of 56 *TaARFs* (excluding 18 splice variants) were identified with relatively conserved motifs within sub-groups. The Ka/Ks ratio of all gene pairs was less than 1, indicating that *TaARF* genes are under negative selection pressure. Gene duplication events were concentrated on the ARFA1 sub-group, suggesting that *TaARFA1* genes are conserved. Gene expression pattern analysis revealed that most *TaARFA1* genes were relatively highly expressed during different growth and development stages, biotic stress, and abiotic stress, indicating that they might play an important role in development and the stress response. TaARFA1 might interact with PP2C, supporting the role of *TaARFA1* genes

in the wheat stress response. Taken together, the present study provided comprehensive insights into the structure, organization, evolution, and expression profiles of the *TaARF* gene family in wheat, which support further functional characterization of *TaARF* family genes for the development of high-quality wheat varieties.

### Funding

This work was supported by the National Key Research and Development Program of China (2017YFD0100802), Hubei Provincial Special Project of Central Government Guides Local Science and Technology Development (2020ZYYD011), the Technology Innovation Project of Hubei Province (2018ABA085), and China Agriculture Research System (CARS-03). The funders had no role in study design, data collection and analysis, decision to publish, or preparation of the manuscript.

### Grant Disclosures

The following grant information was disclosed by the authors:
National Key Research and Development Program of China: 2017YFD0100802.
Hubei Provincial Special Project of Central Government Guides Local Science and Technology Development: 2020ZYYD011.
Technology Innovation Project of Hubei Province: 2018ABA085.
China Agriculture Research System: CARS-03.

### Competing Interests

The authors declare there are no competing interests.

### Author Contributions

- Yaqian Li performed the experiments, analyzed the data, prepared figures and/or tables, authored or reviewed drafts of the paper, and approved the final draft.
- Jinghan Song analyzed the data, authored or reviewed drafts of the paper, and approved the final draft.
- Guang Zhu performed the experiments, authored or reviewed drafts of the paper, and approved the final draft.
- Zehao Hou analyzed the data, prepared figures and/or tables, and approved the final draft.
- Lin Wang and Xiaoxue Wu performed the experiments, prepared figures and/or tables, and approved the final draft.
- Zhengwu Fang and Chunbao Gao conceived and designed the experiments, authored or reviewed drafts of the paper, and approved the final draft.
- Yike Liu conceived and designed the experiments, analyzed the data, authored or reviewed drafts of the paper, and approved the final draft.

### Data Availability

Raw data are available in the Supplementary Files.

## Supplemental Information

Supplemental information for this article can be found online at http://dx.doi.org/10.7717/peerj.10963#supplemental-information.

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
