# Peer review of "Genome-wide identification and expression analysis of ADP-ribosylation factors associated with biotic and abiotic stress in wheat (Triticum aestivum L.)"

_PeerJ, doi:10.7717/peerj.10963_

## Round 0.1 · original submission · Minor Revisions

Dear Authors
The two reviewers have recommended the manuscript after minor revisions. Please pay attention, among suggestions, to the improvement of qRT-PCR data, which represents a very important aspect.
Best Regards
C.A.

Reviewer 1 ·

Basic reporting

English language and style are fine/minor spell check required

It is necessary to improve the resolution of the Figure 2 (A; B and C) and the Figure 5A

Experimental design

No comment

Validity of the findings

no comment

Additional comments

ADP-ribosylation factors (ARFs) belong to Ras superfamily of small GTP-binding proteins (GTPases), which was reported to function in diverse physiological and molecular activities. These low molecular weight (21–24 kDa) proteins was classified into ARFs and ARF-like (ARL) proteins based on their functional characteristics and sequence homology. ARFs are highly conserved proteins (>60% sequence identity), while ARLs are highly divergent (40–60% identity). ARFs was identified in several plant specie (Arabidopsis, rice, tomato, potato, maize, carrot, wheat, tobacco and barley), in which they play a relevant role in a broad-spectrum biological process, including growth and development, senescence and senescence-dependent recycling, tolerance to biotic and abiotic stresses.In the present manuscript, the role of ARF genes was explorated in development of wheat (Triticum aestivum L.). The wheat is one of the most important crops for human, provding staple food globally.
The authors conduced a systematic phylogenetic analysis of TaARFs and analyzed their expression profiles at different development stages and under different biotic and abiotic stresses.74 wheat ARF genes were identified and clustered in 7 sub-groups. TaRFA1 subgroup genes are strongly conserved and induced in response to biotic and abiotic stresses. TaRFA1 proteins seems to interact with protein phosphatase 2 (PPC2), a key protein in the abscisic acid (ABA) signaling patway. This evidence would seem to support the hypothesis that TaRFA1 plays a role in the wheat stress response.In my opinion, the manuscript is interesting and provides comprehensive insights into the structure, organization, evolution, and expression profiles of the TaARF gene family in wheat.
The title is clear and informative about the research carried out. The abstract includes all the main findings reported in the manuscript.The introduction provides sufficient background and include all relevant references. Experimental approach is correct and methods are adequately described. The results are clearly presented, only the resolution of the Figures 2A; 2B and 2C) and the Figure 5A should be improved. The discussion is supported by the results.
English language and style are fine/minor spell check required. Based on my assessment, I am in favour of the publication of the manuscript, whether the authors will provide an accurate revision of the English language and make clear the Figures mentioned above.

Reviewer 2 ·

Basic reporting

The manuscript by Li et al is well written, clear and sufficient background and introduction provided. The manuscript is well structured, the data are solid and raw data provided.

Experimental design

The manuscript is focused on analysis of ADP-ribosylation factors in wheat, one of the most important crops, and provide data and tools for research community to improve crop tolerence to various stresses. The authors made throughout phylogenetic analysis of TaARFs as well as gene structure, location and expression analysis of TaARFs.
Methods are described with sufficient details and information. The results are clearly presented in figures.
One major concern about the experimental design is qRT-PCR data obtained under cold, salt, and ABA stress. According to the MIQE guidelines the use of at least two reference genes is nessesary for a strong and reliable normalisation in RT-qPCR experiments. The authors, however, use only one reference gene in the study. So, it should be performed according to the MIQE standards.

Validity of the findings

no comments

Additional comments

The only weakness of the paper is qRT-PCR data as indicated above, which should be improved before acceptance of the paper.

---

## Round 0.2 · accepted · Accept

Dear Dr. Liu,
Thank you for your submission to PeerJ.

I am writing to inform you that your manuscript - Genome-wide identification and expression analysis of ADP-ribosylation factors associated with biotic and abiotic stress in wheat (Triticum aestivum L.) - has been Accepted for publication. Congratulations!

Congratulations again, and thank you for your submission.

With kind regards,
Carmen Arena
Academic Editor, PeerJ

Reviewer 1 ·

Basic reporting

no comment

Experimental design

no comment

Validity of the findings

no comment

Additional comments

The authors modified the manuscript according to the requests of my first review

Reviewer 2 ·

Basic reporting

no comment

Experimental design

The authors have improved their manuscript by adding one more reference gene for qRT-PCR data normalization. As it was a major concern during first review, I recommend now publication of the paper.

Validity of the findings

no comment

Additional comments

The required additional experiments were done as suggested, the manuscipt can be now published.